# Community's misconception about COVID-19 and its associated factors in Satkhira, Bangladesh: A cross-sectional study

**Md. Bakebillah[1]ⵁ, Md. Arif Billah[2]ⵁ, Befikadu L. Wubishet[3], Md. Nuruzzaman Khan[4]***

**1** Department of Folklore, Faculty of Social Science, Jatiya Kabi Kazi Nazrul Islam University, Trishal, Mymensingh, Bangladesh, **2** Department of Psychology and Counselling, Faculty of Business, Economic and Social Development, Universiti Malaysia Terengganu, Kuala Terengganu, Terengganu, Malaysia, **3** Postdoctoral Research Fellow, Health Services Research Centre, Faculty of Medicine, The University of Queensland, Brisbane, Australia, **4** Department of Population Science, Faculty of Social Science, Jatiya Kabi Kazi Najrul Islam University, Trishal, Mymensingh, Bangladesh

ⵁ These authors contributed equally to this work.
* mdnuruzzaman.khan@uon.edu.au

**Data Availability Statement:** Data is fully available in the body of manuscript as well as in a data respiratory which can be downloaded from here: https://www.openicpsr.org/openicpsr/workspace?

# Abstract

## Introduction

Misconception related to coronavirus disease-2019 (COVID-19) have been spread out broadly and the the World Health Organization declared these as a major challenge to fight against the pandemic. This study aimed to assess COVID-19 related misconception among rural people in Bangladesh and associated socio-demographic and media related factors.

## Methods

Multistage sampling method was used to collect data (n = 210) from three unions of Satkhira District, Bangladesh. The dependent variable was the presence of COVID-19 related misconception (Yes, No) which was generated based on respondents' responses to a set of six questions on various types of misconception. Exposure variables were respondents' socio-demographic characteristics, mass media and social media exposure. Descriptive statistics were used to describe the characteristics of the respondents. Bivariate and multivariate logistic regression models were used to determine the factors associated with COVID-19 misconception.

## Results

More than half of the study respondents had one or more COVID-19 related misconception. Over 50% of the total respondents considered this disease as a punishment from God. Besides, many of the respondents reported that they do not think the virus causing COVID-19 is dangerous (59%) and it is a disease (19%). Around 7% reported they believe the virus is the part of a virus war (7.2%). The bivariate analysis found the presence of socio-demographic factors of the respondents, as well as the factors related to social and mass media, were significantly associated with the COVID-19's misconception. However, once all factors considered together in the multivariate model, misconception were found to be lower among

goToPath=/openicpsr/145501&goToLevel=
project#.

**Funding:** The author(s) received no specific funding for this work.

**Competing interests:** The authors have declared that no competing interests exist.

**Abbreviations:** COVID-19, Coronavirus Disease 2019; AOR, Adjusted Odds Ratio; OR, Odds ratio; SE, Standard Error; CI, Confidence Interval; HIV/AIDS, Human Immuno-deficiency Virus/ Acquired Immunodeficiency Syndrome; WHO, World Health Organization.

secondary (AOR, 0.33, 95% CI: 0.13–0.84) and tertiary (AOR, 0.29, 95% CI: 0.09–0.92) educated respondents compared to the respondents with primary education.

## Conclusion

This study obtained a very higher percentage of misconception about the COVID-19 among the respondents of Satkhira district in Bangladesh. This could be a potential challenge to fight against this pandemic which is now ongoing. Prioritizing mass and social media to disseminate evidence-based information as well as educate people about this disease are necessary.

## Background

Since December 2019, the world has been faced with the coronavirus outbreak which the World Health Organization (WHO) later declared as a pandemic. Approximately 207 million of people have contracted the virus and around 4 million have died due to the disease as of August 13, 2021. Disease severity and fatality is much higher among older people and those with pre-existing chronic diseases such as diabetes and hypertension [1, 2]. Misconception related to the COVID-19 are other important reasons for higher complications and deaths, particularly in the remote regions of low income countries including Bangladesh [3, 4]. For instance, a significant portion of religious people believe that the virus mainly affects the non-religious or atheist [4, 5]. Some people also believe that people in low-income countries have a better immunity than people in high income settings, and therefore the former are less likely to be infected by the COVID-19 [6, 7]. Other groups believe coronavirus is a revenge of nature and punishment from God [8–10]. These sorts of misconception have been spread widely since the onset of the pandemic and similar to what was observed during previous pandemics such as HIV/AIDS and Ebola [11, 12]. Insufficient information about the characteristics of the virus along with the rapidly evolving evidence coming from numerous sources including social media are major reasons for the widespread existence of misconnection [13]. Unfortunately, these misconceptions spread much quicker than evidence-based information through social media and word of mouth. The WHO described this rapid and far-reaching spread of both accurate and inaccurate information as '*infodemic*' [13–15].

Bangladesh is among the worst infected countries with coronavirus. Around 1.4 million people has been infected as of August 13, 2021 and around 24 thousands of which have died according to WHO's estimates for Bangladesh [16]. However, these figures should be underestimated given the very low testing facility and lack of accurate reporting of cases and deaths [17, 18]. Moreover, the widespread existence of misconception and fear of stigma discourages symptomatic individuals from going for testing and reporting cases and deaths. Importantly, these sorts of behaviours are more common among people in rural and remote areas than their counterparts residing in the urban areas [18–20].

There is a lack of study in Bangladesh as well as other low- and lower-middle income countries (LMICs) on the issue of misconception about COVID-19. Only a few studies have been conducted but with several limitations including the inappropriate list of misconception and inadequate sample [21, 22]. Other focus areas of previous studies were knowledge and awareness among the general population, health professionals, parents and students, and vulnerable or susceptible populations [23–30]. Moreover, a very few studies has been conducted to explore misconception about COVID-19 that were disseminated through social and mass media [6, 31, 32]. Evidence from the rural area is scarce till now though around 70% of the

total population in Bangladesh live in rural area and have greater risk of getting infected and deaths given the limited availability of testing and health care facility. Therefore, this study aimed to explore the COVID-19 related misconception among rural people in Bangladesh.

## Methodology

### Data source and sample

Primary data was collected using the multistage sampling during the period April 15 to May 09, 2020. At the first stage of the selection, Satkhira district was selected randomly from the list of 64 districts in Bangladesh. The district consists of 79 unions, of which three unions (Ratanpur, Kusulia and Moutala Union) were selected randomly for this study. These three unions consist 68 villages in total [33] of which a further random sampling was done to select 9 villages, three villages from each union. The target was to include all households in the selected villages. There were 113 households of the selected 9 villages, however, of which 105 households were included finally with the inclusion rate at 92.92%. A fixed number of two members were included from each household with the inclusion criteria, (i) the youngest person aged more than 18 and (ii) the oldest person. These criteria were chosen to present the family's youngest (over age 18) and oldest person. This produced responses from 210 respondents which were our analysis sample for the study (Fig 1).

### Data collection

Data were collected through face-to-face interview with the selected individuals using structured questionnaire. The questionnaire was first prepared in English and translated into

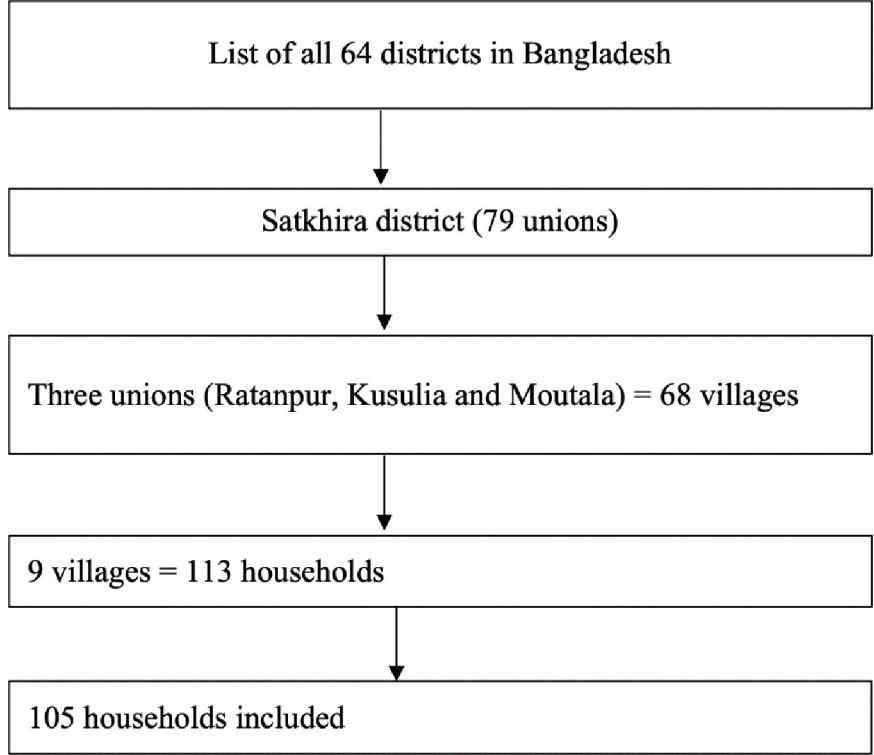

**Fig 1. Sampling procedure.**

Bengali, the official language of Bangladesh. It was then back translated into English by independent translators to make sure the original meaning was retained. The questionnaire was developed based on relevant previous research in Bangladesh and other low-income countries [4, 6, 22, 23, 34, 35]. Two rounds of pre-testing were conducted, and misunderstood questions were revised before finalizing the questionnaire. Ethical review committee of the Institute of Biological Science, University of Rajshahi, reviewed the study proposal and provided ethical approval. All participants were informed of the purpose of the study, the voluntary nature of the study, the anonymous use of the data and informed verbal consent was sought before data collection commenced.

### Dependent variable

The dependent variable was misconception on COVID-19, which was created based on the respondents' response to six types of misconception. These were COVID-19 is: (i) a punishment from God, (ii) revenge of nature, (iii) a virus war, (iv) not a dangerous virus, (v) not a disease and (vi) not a pandemic. Response to each statement was reported dichotomously, "Yes (1)" *or* "No (0)". Aggregating these recoded responses, the misconception variable had been generated with two categories, respondents who had scored greater than and equal to median were considered as having high level of misconception and vice-versa [6].

### Exposure variables

The explanatory variables included were respondents' socio-demographic characteristics and social and mass-media exposure. Socio-demographic characteristics included respondents' gender, age, education and monthly income. Social and mass media exposure related variables included respondents use of social media, daily frequency of social media use, reading newspapers, frequency of reading the newspaper, watching television, frequency of watching television, listening to radio and frequency of listening to the radio.

### Analysis

Descriptive statistics were used to describe the characteristics of the respondents. Chi-square test was used to identify the variables significantly associated with misconception as well as their distribution. Bivariate and multivariate logistic regression models were used to identify the factors associated with COVID-19 related misconception. The variables found significant in the chi-square analysis at 10% level of significance were included in the multivariate logistic regression model. Results were reported as Odd Ratio (OR) with its 95% confidence interval (CI). STATA version 14.3 was used for all statistical analyses.

## Results

### Characteristics of the study population

The characteristics of the study participants are presented in Table 1. Around two-thirds, 139 (66.19%), were males. More than half of the respondents, 113 (53.81%), were middle-aged (30–59 years) and completed their secondary education (43.81%). The mean monthly household income was 15,645 Bangladeshi Taka (BDT, USD 184). More than half of the total respondents (57.61%) lived in a small family and 67.14% of them had children in the family. About 87% of respondents reported use of mass media, of which 41% read newspapers, 82.4% watch television and 6.2% listen to the radio at least once a week. Around 60% of the respondents reported use of social media, more than a quarter of which use social media for more than two hours per day.

**Table 1. Household characteristics of the study population Satkhira district- Bangladesh, 2020.**

| Variables | Mean (SD) | Frequency | Percentage |
|---|---|---|---|
| *Gender* | | | |
| Male | | 139 | 66.19 |
| Female | | 71 | 33.81 |
| *Age group* | 39.46 (16.162) | | |
| <30 years | | 67 | 31.90 |
| 30–59 years | | 113 | 53.81 |
| 60+ years | | 30 | 14.29 |
| *Education* | 9.44 (4.198) | | |
| Primary | | 47 | 22.38 |
| Secondary | | 92 | 43.81 |
| Tertiary | | 71 | 33.81 |
| *Monthly family income* | 15645.24 (10986.25) | | |
| < = 10000 | | 102 | 48.57 |
| >10000 | | 108 | 51.43 |
| *Number of earning person in the family* | | | |
| Only one | | 145 | 69.05 |
| More than one | | 65 | 30.95 |
| *Family member* | 4.55 (1.711) | | |
| Small family (< = 4) | | 121 | 57.62 |
| Extended family | | 89 | 42.38 |
| *Children in the family* | | | |
| No | | 69 | 32.86 |
| Yes | | 141 | 67.14 |
| *Older in the family* | | | |
| No | | 135 | 64.29 |
| Yes | | 75 | 35.71 |

Common misconception among the study population are presented in Table 2. More than half of the total respondents (52.86%) reported one or more misconception. The most popular misconception was, COVID-19 virus is not dangerous (59%) followed by the belief that COVID-19 virus is a punishment from God (47%). Some respondents do not believe that the virus may cause a disease (19%) and some believe that COVID-19 is a virus war (7.6%).

The distribution of respondents with or without misconception across socio-demographic and media related factors are presented in Table 3. Misconception were reported by higher proportion of men (74.65%) than women (25.25%). Total of 78 (70.27%) respondents who do not read newspaper had a high level of misconception whereas 87 (87.88%) of respondents who watched television had a low level of misconception about COVID-19. Besides, higher percentage of respondents who use social media (72.73%) reported misconception regarding COVID-19 compared to those who reported not using social media (52.25%). The study also found that gender, monthly income, television use, treatment facilities, remedy measures were significantly associated with COVID-19 related misconception (p<0.05) in the bivariate analysis.

Results from the unadjusted and adjusted multivariate logistic regression analyses are presented in Table 4. The unadjusted logistic analysis found a higher likelihood of misconception among women (OR, 2.1, 95% CI: 1.16–3.77) compared with men. Respondents who had reported their monthly income as more than 10000 BDT were 46% (OR, 0.54, 95% CI: 0.31–

**Table 2. Common COVID-19 related misconception among rural people in Satkhira district- Bangladesh, 2020.**

| Attributes | Frequency | Percentage |
|---|---|---|
| COVID-19 is a punishment from God | | |
| Have no misconception | 111 | 52.9 |
| Have misconception | 99 | 47.1 |
| COVID-19 is the revenge of nature | | |
| Have no misconception | 205 | 97.6 |
| Have misconception | 5 | 2.4 |
| COVID-19 is a virus war | | |
| Have no misconception | 194 | 92.4 |
| Have misconception | 16 | 7.6 |
| COVID-19 is a dangerous virus | | |
| Have no misconception | 87 | 41.1 |
| Have misconception | 123 | 58.6 |
| COVID-19 is a disease | | |
| Have no misconception | 171 | 81.4 |
| Have misconception | 39 | 18.6 |
| COVID-19 is a pandemic | | |
| Have no misconception | 209 | 99.5 |
| Have misconception | 1 | 0.5 |

0.93) less likely to have misconception compared with those who reported their monthly income to be less than 10000 BDT (p<0.01). Compared with individuals who had primary education, those who attained secondary and tertiary level of education were 74% (OR, 0.26, 95% CI: 0.12–0.58) and 81% (OR, 0.19, 95%CI: 0.08–0.43) less likely to have misconception, respectively. Besides, the use of mass media (OR, 0.26, 95%: 0.10–67), and social media for ≤2 hours per day (OR, 0.40, 95% CI: 0.21–0.78) and >2 hours per day (OR, 0.28, 95% CI: 0.14–0.57) were found to be associated lower level of COVID-19 related misconception.

Adjusting for confounders, only education was found to be significantly associated with having COVID-19 misconception. Respondents with secondary and tertiary education were found to be 67% (AOR,0.33, 95% CI: 0.13–0.84) and 71% (AOR, 0.29, 95% CI: 0.09–0.92) less likely to report misconception about COVID-19, respectively, compared to those with primary education.

## Discussion

This study identified common misconception related to COVID-19 in Bangladesh and their association with factors related to sociodemographic, mass and social media use of respondents. COVID-19 related misconception was found with significant variation across the respondents' socio-demographic and media related factors. Higher level of education was found to be significantly related to the lower level of misconception about COVID-19 among rural Bangladesh people. These findings are robust and could have an important policy implication to reduce misconception on this pandemic in Bangladesh and other LMICs.

More than half (52%) of the study participants reported a very high level of misconception. The commonest forms of misconception included considering the pandemic as a punishment of nature and the coronavirus is not a dangerous virus. These types of misconception were promoted by religious leaders and covered in several social and mass media [5, 9]. Moreover, though it is established that the COVID-19 is a spill over infection and not a human-made (laboratory-generated) virus [36–38], another common misconception is that this virus is

**Table 3. Socio-demographic and media related variables among respondents with level of COVID-19 related misconception, Satkhira district- Bangladesh, 2020.**

| Variable | COVID-19 related misconception | | Chi-square (df), p-value |
|---|---|---|---|
| | Low | High | |
| *Gender* | | | *6.1286 (1), 0.013* |
| Male | 74 (74.65) | 65 (58.56) | |
| Female | 25 (25.25) | 46 (41.44) | |
| *Education* | | | *17.3052 (2), 0.000* |
| Primary | 10 (1.11) | 37 (33.33) | |
| Secondary | 47 (47.47) | 45 (40.54) | |
| Tertiary | 42 (42.42) | 29 (26.13) | |
| *Monthly family income* | | | *5.0016 (1), 0.025* |
| < = 10000 | 40 (40.40) | 62 (55.86) | |
| >10000 | 59 (59.60) | 49 (44.14) | |
| *Read Newspaper* | | | *11.2862 (1), 0.001* |
| No | 47 (47.47) | 78 (70.27) | |
| Yes | 52 (52.53) | 33 (29.73) | |
| *Watch Television* | | | *3.9004 (1), 0.048* |
| No | 12 (12.12) | 25 (22.25) | |
| Yes | 87 (87.88) | 86 (77.48) | |
| *Listen Radio* | | | *2.7132 (1), 0.100* |
| No | 90 (90.91) | 107 (96.40) | |
| Yes | 9 (9.09) | 4 (3.60) | |
| *Frequency of reading newspaper* | | | *10.6437 (2), 0.005* |
| Not read | 46 (46.46) | 76 (68.47) | |
| Read but not daily | 13 (13.13) | 7 (6.31) | |
| Read daily | 40 (40.40) | 28 (25.23) | |
| *Frequency of watching Television* | | | *4.9216 (2), 0.085* |
| Not watch | 12 (12.12) | 26 (23.42) | |
| Watch but not daily | 5 (5.05) | 7 (6.31) | |
| Watch daily | 82 (82.83) | 78 (70.27) | |
| *Frequency of listening radio* | | | *3.0914 (2), 0.213* |
| Not listen | 90 (90.91) | 107 (96.40) | |
| Listen but not daily | 5 (5.05) | 3 (2.70) | |
| Listen daily | 4 (4.04) | 1 (0.90) | |
| *Use social media* | | | *13.5524 (1), 0.000* |
| No | 27 (27.27) | 58 (52.25) | |
| Yes | 72 (72.73) | 53 (47.75) | |
| *Frequency of using social media/ day* | | | *14.4197 (2), 0.001* |
| Don't use | 27 (27.27) | 58 (52.25) | |
| < = 2 hrs | 36 (36.36) | 31 (27.93) | |
| >2 hrs | 36 (36.36) | 22 (19.82) | |
| *Treatment facility* | | | *7.5754 (2), 0.023* |
| No treatment | 45 (45.45) | 53 (47.75) | |
| Non-specialized treatment | 16 (16.16) | 32 (28.83) | |
| Specialized treatment | 38 (38.38) | 26 (23.42) | |
| *Remedy* | | | *6.8192 (2), 0.033* |
| Prayer | 7 (7.07) | 14 (12.61) | |
| Awareness | 41 (41.41) | 28 (25.23) | |

*(Continued)*

**Table 3.** (Continued)

| Variable | COVID-19 related misconception | | Chi-square (df), p-value |
|:---:|:---:|:---:|:---:|
| | **Low** | **High** | |
| Both | 51 (51.52) | 69 (62.16) | |
| *Role of social media* | | | *16.5407 (2), 0.000* |
| Negative | 8 (8.08) | 6 (5.41) | |
| Positive | 78 (78.79) | 63 (56.76) | |
| NA | 13 (13.13) | 42 (37.84) | |

human-created biological weapon for political reason [39]. Similar findings were reported across the world, including America, Saudi Arabia, Nigeria, Nepal, Ghana, Uganda and Pakistan [8, 10, 32, 34, 40–44].

The socio-demographic characteristics of the respondents were associated with the COVID-19 related misconception. For instance, COVID-19 related misconception are more common among people living in rural areas as reported in a previous study in Bangladesh [23]. People living in rural areas usually have lower education and have limited access to healthcare facilities and mass media that could contribute to the more widespread presence of such misconception. This is particularly true for the rural women of Bangladesh, as they are even more family-centric and most have lower education which is usually associated with higher infectious disease burden and related misconception. For instance, a significant proportion of rural women believe children's respiratory disease (acute respiratory disease) was the influence of evils or supernatural beings, and therefore can only be treated through the spiritual healers [45, 46]. Though these sorts of misconception are different from the misconception related to the COVID-19, their existence creates a fertile ground for COVID-19 related misconception.

The Government of Bangladesh is now using mass media to reduce misconception about the ongoing COVID-19 pandemic and disseminate correct information. The WHO recommends the same strategy to fight against this pandemic. Newspapers and television channels have been broadcasting regular updates about the spread of the disease, new findings, national and international strategies to combat the pandemic. However, access to mass media, except the radio, is significantly lower in the rural area of Bangladesh compared to the urban area. However, the effectiveness of the radio to disseminate such information has been questioned for a variety of reasons. For instance, radio often disseminates one way of information, which is most difficult to understand for the rural and uneducated people. Moreover, such one-way communication can have little influence to justify and reduce misconception that the rural people get from other sources including social media where unauthentic information spread out rapidly. Additionally, it mostly fails to answer the questions that people have had regarding this disease, thorough this is an important way out to reduce misconception. Ensuring proper access to mass media and dissemination of correct knowledge could therefore have a significant role to reduce misconception about the COVID-19 pandemic.

This study has several strengths and limitations. To the best of available literature, this is the first study of its kind in Bangladesh that assess the misconception regarding the COVID-19 pandemic and associated factors. Moreover, this study considers a wide range of factors and identify their association with the COVID-19 related misconception. This makes the study findings more reliable. However, the data collected was cross-sectional in nature and therefore these studies findings are associations only, not necessarily causal. Besides, the data was collected during the first wave of COVID-19 in Bangladesh, and therefore the findings are

**Table 4. Factors associated with the misconception about COVID-19, Satkhira district- Bangladesh, 2020.**

| Variables | Misconception of COVID-19 | | | |
|---|---|---|---|---|
| | OR (SE) | 95% CI | AOR (SE) | 95% CI |
| *Gender* | | | | |
| Male | 1 | | 1 | |
| Female | 2.10 (0.631)** | 1.1611–3.7791 | 1.64 (0.569) | 0.8339–3.2376 |
| *Age group* | | | | |
| <30 years | 1 | | 1 | |
| 30–59 years | 0.89 (0.274) | 0.4829–1.6251 | 0.57 (0.216) | 0.2738–1.2008 |
| 60+ years | 0.81 (0.357) | 0.3422–1.9211 | 0.56 (0.311) | 0.1890–1.6643 |
| *Education* | | | | |
| Primary | 1 | | 1 | |
| Secondary | 0.26 (0.107)*** | 0.1152–0.5813 | 0.33 (0.157)* | 0.1289–0.8398 |
| Tertiary | 0.19 (0.0803)*** | 0.0803–0.4339 | 0.29 (0.171)* | 0.0924–0.9171 |
| *Monthly income* | | | | |
| < = 10000 | 1 | | 1 | |
| >10000 | 0.54 (0.150)** | 0.3094–0.9279 | 0.85 (0.288) | 0.4347–1.6466 |
| *Use of Mass media* | | | | |
| No | 1 | | 1 | |
| Yes | 0.26 (0.126)** | 0.1011–0.6737 | 0.77 (0.378) | 0.2372–2.5120 |
| *Role of Mass media* | | | | |
| Negative | 1 | | 1 | |
| Positive | 0.89 (0.510) | 0.2866–2.7393 | 0.61 (0.411) | 0.1626–2.2849 |
| NA/Others | 2.57 (2.062) | 0.5342–12.3781 | 0.75 (0.733) | 0.1105–5.0945 |
| *Use of Social media* | | | | |
| No use | 1 | | 1 | |
| < = 2 hrs per day | 0.40 (0.136)** | 0.2066–0.7777 | 0.97 (0.453) | 0.3893–2.4217 |
| >2 hrs per day | 0.28 (0.102)*** | 0.1413–0.5728 | 0.70 (0.378) | 0.2495–2.0165 |
| *Role of social media* | | | | |
| Negative | 1 | | 1 | |
| Positive | 1.08 (0.610) | 0.3551–3.2656 | 1.46 (0.961) | 0.4000–5.3053 |
| NA/others | 4.31 (2.698)* | 1.2619–14.7046 | 3.23 (2.438) | 0.7372–14.1760 |
| *Treatment facility* | | | | |
| No treatment | 1 | | 1 | |
| Non-specialized treatment | 1.70 (0.624) | 0.8268–3.4877 | 1.52 (0.642) | 0.6608–3.4762 |
| Specialized treatment | 0.58 (0.189)* | 0.3070–1.0992 | 0.94 (0.354) | 0.4519–1.9691 |
| *Remedy* | | | | |
| Prayer | 1 | | 1 | |
| Awareness | 0.34 (0.179)** | 0.1223–0.9533 | 0.55 (0.323) | 0.1731–1.7394 |
| Both | 0.68 (0.337) | 0.2547–1.7967 | 0.88 (0.489) | 0.2957–2.6134 |

Log likelihood = -124.85775, LR chi-square (17) = 40.72, p = 0.0010 and Pseudo R-square = 0.1402

Note that-

*p<0.05

**p<0.01

***p<0.001. N = 210.

representative of the misconception that were common at the first wave of COVID-19. However, these study findings are still valid and important for policy implications. Moreover, information was collected only from one district and three unions and therefore the findings may

not be generalizable to the whole population of Bangladesh through the district and unions included were selected randomly.

## Conclusion

Misconception about the COVID-19 disease are significantly prevalent among the rural people in Satkhira, Bangladesh. Respondents' socio-demographic characteristics and mass and social media usage behaviours were associated with COVID-19 related misconception. Proper education and dissemination of correct information through mass and social media are important to reduce misconception and to succeed in the fight against this pandemic.

## Acknowledgments

We acknowledged the respondents who had participated in this study voluntarily. We also give thanks to Mst. Sumaiya Sultana (Post-graduate student), Sk Mustafizur Rahman (physiotherapist), Prokash Patra (Science teacher at a high school), Ajgor Ali (Post-graduate student), and Jayanta Kumar Ghosh (Village doctor) for being associated during the data collection at the field level. We also acknowledged the Department of Folklore, Faculty of Social Science, Jatiya Kabi Nazrul Islam University, Trishal-2224, Mymensingh, Bangladesh where this study was conducted.

## Author Contributions

**Conceptualization:** Md. Bakebillah, Md. Nuruzzaman Khan.

**Data curation:** Md. Bakebillah.

**Formal analysis:** Md. Arif Billah.

**Methodology:** Md. Nuruzzaman Khan.

**Supervision:** Md. Nuruzzaman Khan.

**Writing – original draft:** Md. Bakebillah, Md. Arif Billah.

**Writing – review & editing:** Md. Arif Billah, Befikadu L. Wubishet, Md. Nuruzzaman Khan.

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
