## [Decision Letter · Decision Letter 0]

26 May 2021

PONE-D-21-10887

Community’s misconception about COVID-19 and its associated factors: Evidence from a cross-sectional study in Bangladesh

PLOS ONE

Dear Dr. Khan,

Thank you for submitting your manuscript to PLOS ONE. After careful consideration, we feel that it has merit but does not fully meet PLOS ONE’s publication criteria as it currently stands. Therefore, we invite you to submit a revised version of the manuscript that addresses the points raised during the review process.

This study was conducted based on a primary data, however no ethics approval report were provided. Without ethics approval, the analysis results cannot be published. Also there were several issues identified by the reviewers those need to be fixed before taking final decision.  

We look forward to receiving your revised manuscript.

Kind regards,

Enamul Kabir

Academic Editor

PLOS ONE

Journal Requirements:

2. PLOS ONE does not copy edit accepted manuscripts (https://journals.plos.org/plosone/s/criteria-for-publication#loc-5). To that effect, please ensure that your submission is free of typos and grammatical errors.

3. During our internal evaluation of the manuscript, we found some text and structural overlap between your submission and the following previously published works:

- Mekonnen, H.S., Azagew, A.W., Wubneh, C.A. et al. Community’s misconception about COVID-19 and its associated factors among Gondar town residents, Northwest Ethiopia. Trop Med Health 48, 99 (2020). https://doi.org/10.1186/s41182-020-00279-8

We would like to make you aware that copying extracts from previous publications, including the title, word-for-word is unacceptable. In addition, the reproduction of text from published reports has implications for the copyright that may apply to the publications.

Please revise the manuscript to rephrase the duplicated text.

Reviewers' comments:

Reviewer's Responses to Questions

**Comments to the Author**

1. Is the manuscript technically sound, and do the data support the conclusions?

Reviewer #1: No

Reviewer #2: Yes

2. Has the statistical analysis been performed appropriately and rigorously? 

Reviewer #1: No

Reviewer #2: Yes

3. Have the authors made all data underlying the findings in their manuscript fully available?

Reviewer #1: Yes

Reviewer #2: Yes

4. Is the manuscript presented in an intelligible fashion and written in standard English?

Reviewer #1: Yes

Reviewer #2: Yes

5. Review Comments to the Author

Reviewer #1: Overall

1. Write up and language need to revisit and proof read adequately by authors, if needed advised to take assistance from native speakers

2. Authors methodology is not fully clear and replicable

3. Some of the result sections are not very clear and some selection seems biased on authors judgement

Specific

1. Section, Bakcground, P1, L5, check punctuation error “pandemic. [1].”

2. Section, Bakcground, P1, Please update the data based on current report “Bangladesh is one of the countries worst hit by the COVID-19 pandemic. More than a half million people has been infected among which around 9,000 people were died as per the official estimates of the World Health Organization for Bangladesh [15].”

3. Method section “Primary data was collected during the period April 15 to May 09, 2020.” Which indicates study has been carried out long back during initial phase of pandemic and now already peak of second wave is ongoing so these findings may be not fully applicable to current scenario, which need to be discussed in limitation regarding cautious interpretation of the findings.

4. Method section, Data collection subsection: why not written consent?? “Verbal consent was taken from each respondent before collecting the data.”

5. Method section, Dependent variable, “These were COVID-19 is: (i) a punishment from god, (ii) revenge of nature, (iii) a virus war, (iv) not a dangerous virus, (v) not a disease and (vi) not a pandemic.” Were these six only misconceptions there?? If so what is the source? IF nor why these were only taken for dependent variables?

6. Methods section: I could see no mention of ethical consideration and their details in the methods and how verbal consent is enough for this study?

7. Result section: Table 1, arbitrarily while reporting normally distributed data, it is supposed to use Mean(±SD) than Mean (SE), please check and do needful

8. Result section: Table 3, how misconception present or absent decided? There are six questions about misconceptions in table 2 and it is divided into binary variable in table 3. I could not get how it is derived? Seems like punishment from god taken as misconception for table 3, but it’s labeling is opposite than in table 2. If so based on what rest attributes were neglected?

Reviewer #2: upon checking your manuscript" " i found that the manuscript is very important and timely issues that contribute for the tackling of the pandemic of COVID-19/ that helps the prevention of COVID-19 pandemic through identifying the community misconception to propose future plan.

Constructive Comments for Author(s)

However, the following points need to be answered/addressed. In order to make your manuscript to the standards of scientific community in sharing knowledge and experiences of your work with other scientific or researchers the following points are forwarded.

The topic “Community’s misconception about COVID-19 and its associated factors: Evidence from a cross-sectional study in Bangladesh”

Better to make like this “Community’s misconception about COVID-19 and associated factors in Satkhira, Bangladesh: a cross-sectional study”

Check grammars/spellings error carefully

On abstract methods part elaborate how you make selection of your sample

COVID-19: write in full words for the first time particularly in the Abstract

Mentions the level of confidences and measure of association (95% CI: AOR/OR)

Multistage sampling was employed why not use design effect?

On justification for conducting the study you state “lack of study in Bangladesh on the issue of misconception about COVID-19” add other reasons while this not enough.

Introduction: re writes and limits to 2 pages the introduction

Operational definition for Community’s misconception: give clear definition based on your classification categories after using ‘PCA’ not on each statement(s) response. Consult how to do PCA/computing different variables to generate those having and/not having misconception. What is your cut off points to say this % of individuals and over all scores should be computed to classify as _or + /have and/do not have misconception.

Which one you’re using median or mean score? Write clearly on this part

Result

Result: put the number then in bracket the %, like this No (%)

“Table 1. Household characteristics of the study population” give full name for the table (time and place)

Other focus areas of previous studies have been knowledge and awareness about COVID-19 among the general population [22], health professionals [23, 24], parents [25] and students [26] and vulnerable and susceptible populations [27–29].rewrite these paragraph put the reference in single bracket it’s from page 3

Look at On page 10 “ Table 4. Unadjusted logistic analysis found higher likelihood of misconception among women (OR, 2.1, 95% CI, 1.16 - 3.77) compared with men.” Explain in this way Unadjusted logistic analysis found that the likelihood of misconception among women is 2times higher than men counterpart(OR, 2.1, 95% CI, 1.16 - 3.77).

Unadjusted logistic analysis found that the likelihood of misconception among women is 2 times higher when compared with men (OR, 2.1, 95% CI, 1.16 - 3.77).

Figures should be attached at the end of document

Discussion

Similar findings were reported across the world, including America [9, 39], Saudi Arabia [31] Nigeria [33,40], Nepal [7], Ghana [41], Uganda [42] and Pakistan [43].rewrite and put references in single bracket

Better if you avoid ‘we’ and use other terms like researchers ….in discussion mention study that also oppose /against your study finding, don’t only use the supportive one.

Conclusion:

Moreover, this study data was collected from only one district and three unions, therefore the findings may not be geralizable for the whole population of Bangladesh though the district and unions included were selected randomly.

Abstract: Conclusion we found that many people living in rural parts of Bangladesh have one or more misconceptions about COVID-19.we obtained a very higher percentage of the rural people of Bangladesh having one or more misconceptions related to COVID-19. Why you conclude for rural people of Bangladesh? Your study was from 210 respondents from three unions of Satkhira District out of 78 unions or 1distric out of 68 districts. So, reconsider your conclusions and make narrow the scope particular to your target populations/target geographic area.

6. PLOS authors have the option to publish the peer review history of their article (what does this mean?). If published, this will include your full peer review and any attached files.

Reviewer #1: No

Reviewer #2: No

---

## [Author Response · Author response to Decision Letter 0]

20 Jul 2021

A MS word file (entitled response to the reviewers) has been added where we have provided point by point response for every reviewers' comments.

---

## [Decision Letter · Decision Letter 1]

10 Aug 2021

PONE-D-21-10887R1

Community’s misconception about COVID-19 and its associated factors in Satkhira, Bangladesh: a cross-sectional study

PLOS ONE

Dear Dr. Khan,

Thank you for submitting your manuscript to PLOS ONE. After careful consideration, we feel that it has merit but does not fully meet PLOS ONE’s publication criteria as it currently stands. Therefore, we invite you to submit a revised version of the manuscript that addresses the points raised during the review process.

We look forward to receiving your revised manuscript.

Kind regards,

Enamul Kabir

Academic Editor

PLOS ONE

Journal Requirements:

Additional Editor Comments (if provided):

Reviewers' comments:

Reviewer's Responses to Questions

**Comments to the Author**

1. If the authors have adequately addressed your comments raised in a previous round of review and you feel that this manuscript is now acceptable for publication, you may indicate that here to bypass the “Comments to the Author” section, enter your conflict of interest statement in the “Confidential to Editor” section, and submit your "Accept" recommendation.

Reviewer #1: All comments have been addressed

Reviewer #2: All comments have been addressed

2. Is the manuscript technically sound, and do the data support the conclusions?

Reviewer #1: Yes

Reviewer #2: Yes

3. Has the statistical analysis been performed appropriately and rigorously? 

Reviewer #1: Yes

Reviewer #2: Yes

4. Have the authors made all data underlying the findings in their manuscript fully available?

Reviewer #1: Yes

Reviewer #2: Yes

5. Is the manuscript presented in an intelligible fashion and written in standard English?

Reviewer #1: Yes

Reviewer #2: Yes

6. Review Comments to the Author

Reviewer #1: Many issues seems to be addressed, however labelling of table still has MD(SE), instead I assume that need to be MD(SD) as the value has changed!

Rest may be ok, however for potential errors as mentioned earlier, it is advised to proof read and finalize for needful ached.

Reviewer #2: 28/07/2021

Constructive Comments on Revised manuscript for Author(s) response

Dear /sir, nice to hear from you. So far most of the Comments given for you previously were addressed/ responded correctly as expected from you that are great. I have few points to be corrected in order to make your manuscript water proof before publication is commenced. So, I’ll appreciate for your concern with regard to the current comments which was given below:-

The primary topic “Community’s misconception about COVID-19 and its associated factors: Evidence from a cross-sectional study in Bangladesh” the manuscript title was revised as it was suggested.

Community’s misconception about COVID-19 and its associated factors in Satkhira, Bangladesh: a cross-sectional study

Result: in writing up the result part write the number then in the bracket write the percentage (%), like this No (%) e.g. This is incorrect ‘Most of the respondents (113, 53.81%)’ should be written as more than half of the respondents 113(53.81%)

With regard to the comments given about the topic was well addressed that is fine.

New comment Abstract part on methodology the last line particularly need to correct it: Univariate and multivariate logistic regression models were used to determine the factors associated with COVID-19 misconception 0.33 (95% CI: 0.13-0.84)... Is it Univar ate or Bivariate & multivariate used to determine associated factors? Check and correct it please.

Introduction and methodology parts were well addressed as per the comments given to them

Introduction parts were written to less than 2page, and multistage sampling was stated briefly

Response to each statement was reported dichotomously, “Yes (1)” or “No

(0)”. Aggregating these recoded responses, the misconception variable had been generated with

two categories, respondents who had scored greater than and equal to median were considered having high level of misconception and vice-versa. The scoring was stated clearly well addressed

Conclusion: The summary was concluded /corrected as the provided comments but add the new comment.

This study obtained a very higher percentage of misconception about the COVID-19 among the respondents analysed(of Satkhira district in Bangladesh.) . This could be a potential challenge to fight against this pandemic which is now ongoing. Prioritizing mass and social media to disseminate evidence-based information as well as educate people about this disease is necessary.

Results

Around two-thirds (139, 66.19%) were males .replace with this do like this 139(66.19%).This is incorrect 'Most of the respondents (113, 53.81%) 'write this as More than half of 113(53.8%)

On Page 8, Table 3. Misconception were reported by higher proportion of men (74, 74.65%) than women (25, 25.25%). 70.27% (78) of the respondents who do not read Newspaper had a high level of misconception whereas 87.88% (87) of…correct as the e.g. given here for all upon writing the results. New comment:-

Table 1. Household characteristics of the study population in three villages of Satkhira , 2020

Table 1. Household characteristics of the study population in Satkhira district - Bangladesh, 2020 (Better if you replace with this

Discussion

e.g. Similar findings were reported across the world, Including America, Saudi Arabia, Nigeria, Nepal, Ghana, Uganda and Pakistan [8, 10, 32, 34, 40–44]. The comments given about discussion were correctly responded. New comment:-Check reference No 33

7. PLOS authors have the option to publish the peer review history of their article (what does this mean?). If published, this will include your full peer review and any attached files.

Reviewer #1: No

Reviewer #2: No

---

## [Author Response · Author response to Decision Letter 1]

13 Aug 2021

We have added a file named "Response to reviewers" where we address reviewers' all comments. Please have a look.

---

## [Decision Letter · Decision Letter 2]

1 Sep 2021

Community’s misconception about COVID-19 and its associated factors in Satkhira, Bangladesh: a cross-sectional study

PONE-D-21-10887R2

Dear Dr. Khan,

We’re pleased to inform you that your manuscript has been judged scientifically suitable for publication and will be formally accepted for publication once it meets all outstanding technical requirements.

Kind regards,

Enamul Kabir

Academic Editor

PLOS ONE

Additional Editor Comments (optional):

Reviewers' comments:

Reviewer's Responses to Questions

**Comments to the Author**

1. If the authors have adequately addressed your comments raised in a previous round of review and you feel that this manuscript is now acceptable for publication, you may indicate that here to bypass the “Comments to the Author” section, enter your conflict of interest statement in the “Confidential to Editor” section, and submit your "Accept" recommendation.

Reviewer #1: All comments have been addressed

Reviewer #2: All comments have been addressed

2. Is the manuscript technically sound, and do the data support the conclusions?

Reviewer #1: Partly

Reviewer #2: Yes

3. Has the statistical analysis been performed appropriately and rigorously? 

Reviewer #1: Yes

Reviewer #2: Yes

4. Have the authors made all data underlying the findings in their manuscript fully available?

Reviewer #1: Yes

Reviewer #2: Yes

5. Is the manuscript presented in an intelligible fashion and written in standard English?

Reviewer #1: Yes

Reviewer #2: Yes

6. Review Comments to the Author

Reviewer #1: In this revision authors seems to revise and proof read with needful corrections making their manuscript readable and comprehend.

Reviewer #2: 8/19/2021

Comments given for authors

Dear authors, is everything ok? I hope this message find you well. Nice to hear such

Very nice work from you hence did /you addressed almost all comments given for you except this two No (%) ,the green color shows the correct ways of writing continue for others,too.and uniform throughout the document.

In result writing please address others too, I have a concern that it would be better to put the number out of bracket and percentage in the bracket like the following Examples: COVID -19 is a curable disease 369 (93.4 %) and COVID-19 cause death Can 355 (89.9%)

See this in the document

The characteristics of the study participants are presented in Table 1. Around two-thirds, 139

(66.19%) were males. More than half of the respondents, 113 (53.81%), were middle-aged (30-

59 years) and completed their secondary education (43.81%). The mean monthly household

income was 15,645 Bangladeshi Taka (BDT, USD 184). More than half of the total respondents

(57.61%) lived in a small family and 67.14% of them had children in the family. About 87% of

respondents reported use of mass media, of which 41% read newspapers, 82.4% watch television

and 6.2% listen to the radio at least once a week. Around 60% of the res… of men (74.65%) than women (25.25%). Total of 78 (70.27%) respondents who do not read newspaper had a high level of misconception whereas 87 (87.88%) of respondents who watched television had a low level of misconception about COVID-19. Besides, higher percentage of respondents who use social media (72.73%) reported misconception regarding COVID-19 compared to those who reported not using social media (52.25%). The study also found that

‘Correct reference 33’ it is not readable, what is all this numbers?

সাতক্ষীরা জেলা. [Internet]. 2021 Apr 23[cited 20 Apr 2021]. Available from:

http://www.satkhira.gov.bd/site/page/23cf9586-1c4b-11e7-8f57-

19

286ed488c766/%E0%A6%8F%E0%A6%95%20%E0%A6%A8%E0%A6%9C%E0%A6%B0%E0%

A7%87%20%E0%A6%B8%E0%A6%BE%E0%A6%A4%E0%A6%95%E0%A7%8D%E0%A6%B7

%E0%A7%80%E0%A6%B0%E0%A6%BE%20%E0%A6%9C%E0%A7%87%E0%A6%B2%E0%A6

%BE

7. PLOS authors have the option to publish the peer review history of their article (what does this mean?). If published, this will include your full peer review and any attached files.

Reviewer #1: No

Reviewer #2: **Yes: **Junayde Abdurahmen Ahmed

---

## [Editor Report · Acceptance letter]

3 Sep 2021

PONE-D-21-10887R2 

Community’s misconception about COVID-19 and its associated factors in Satkhira, Bangladesh: a cross-sectional study 

Dear Dr. Khan:

I'm pleased to inform you that your manuscript has been deemed suitable for publication in PLOS ONE. Congratulations! Your manuscript is now with our production department. 

Kind regards, 

on behalf of

Dr. Enamul Kabir 

Academic Editor

PLOS ONE